# Human Occupancy Monitoring and Positioning with Speed-Responsive Adaptive Sliding Window Using an Infrared Thermal Array Sensor

**DOI:** 10.3390/s25010129

**Published:** 2024-12-28

**Authors:** Yukai Lin, Qiangfu Zhao

**Affiliations:** Department of Computer and Information Systems, The University of Aizu, Aizuwakamatsu 965-8580, Fukushima, Japan; qf-zhao@u-aizu.ac.jp

**Keywords:** occupancy monitoring, indoor positioning, infrared thermal array sensor, adaptive threshold

## Abstract

In the current era of advanced IoT technology, human occupancy monitoring and positioning technology is widely used in various scenarios. For example, it can optimize passenger flow in public transportation systems, enhance safety in large shopping malls, and adjust smart home devices based on the location and number of occupants for energy savings. Additionally, in homes requiring special care, it can provide timely assistance. However, this technology faces limitations such as privacy concerns, environmental factors, and costs. Traditional cameras may not effectively address these issues, but infrared thermal sensors can offer similar applications while overcoming these challenges. Infrared thermal sensors detect the infrared heat emitted by the human body, protecting privacy and functioning effectively day and night with low power consumption, making them ideal for continuous monitoring scenarios like security systems or elderly care. In this study, we propose a system using the AMG8833, an 8 × 8 Infrared Thermal Array Sensor. The sensor data are processed through interpolation, adaptive thresholding, and blob detection, and the merged human heat signatures are separated. To enhance stability in human position estimation, a dynamic sliding window adjusts its size based on movement speed, effectively handling environmental changes and uncertainties.

## 1. Introduction

In recent years, the rapid advancement of the Internet of Things (IoT) has facilitated the development of various smart technologies that are increasingly being integrated into everyday environments. Among these, Human Occupancy Monitoring and Positioning systems have garnered significant attention due to their potential to enhance public safety, optimize resource usage, and improve the quality of life. These systems are particularly valuable in smart cities, where they can be employed to monitor and manage passenger flow in public transportation, enhance security in large commercial spaces, and contribute to the efficient operation of smart homes.

Occupancy monitoring plays a critical role in energy conservation, as demonstrated by various studies. Ref. [1] highlights how combining video data with CO_2_ concentration measurements can effectively detect the number of occupants in an indoor space. These data are then used to control air conditioning (AC), outdoor air handling units (OAHU), and lighting systems, leading to more efficient energy use. Ref. [2] shows how the analysis of office surveillance videos to detect occupancy can contribute significantly to energy savings in building management.

In addition to these specific cases, broader reviews such as [3,4] provide a comprehensive overview of various techniques used for the estimation and detection of building occupancy. Numerous studies have explored different technologies for estimating human occupancy in buildings, including cameras, Wi-Fi, CO_2_ sensors, and electricity sensors.

Despite their broad applications, traditional human occupancy monitoring systems often rely on optical cameras, which pose significant privacy concerns. In addition, these systems may perform poorly in low-light conditions, limiting their effectiveness. To address these issues, many existing approaches have adopted passive infrared (PIR) sensors as alternatives to traditional cameras. For instance, refs. [5,6] have demonstrated the effectiveness of PIR sensors in occupancy estimation. Although PIR sensors offer certain advantages, such as lower cost and reduced privacy concerns, they also have notable limitations. For example, PIR sensors can be prone to detection errors due to external factors, particularly in extreme temperature conditions, where their sensitivity may decrease. They also depend on motion to trigger detection, rendering them unable to detect stationary objects.

In contrast, infrared thermal sensors detect the heat emitted by the human body by sensing infrared radiation from the surface of objects. These sensors can detect objects even when they are stationary, making them a compelling alternative. Infrared thermal sensors operate effectively both during the day and at night, providing reliable detection without compromising privacy. Furthermore, their relatively low power consumption makes them ideal for continuous monitoring applications.

The AMG8833 infrared thermal sensor [7] is an example of such a technology. This sensor is an 8 × 8 infrared thermal array capable of detecting temperature changes with a resolution sufficient to monitor human presence and movement. However, the relatively low resolution of the sensor presents challenges in accurately distinguishing between multiple individuals and precisely determining their positions dynamically.

To address these challenges, this study proposes a Human Occupancy Monitoring and Positioning system that utilizes a combination of several image processing and adaptive thresholding techniques. The core of this system is an improved adaptive thresholding method that dynamically adjusts to varying thermal patterns, ensuring accurate foreground segmentation and reducing the influence of background noise. Additionally, the system integrates a speed-responsive adaptive sliding window as an auxiliary mechanism to further stabilize the detected positions during sudden movement changes. This approach allows the system to maintain both accuracy and robustness when tracking human positions in indoor environments.

The proposed system not only improves the accuracy of body position estimation but also enhances robustness in dealing with uncertainties and fluctuations common in real-world environments. To achieve continuous tracking, the system leverages a combination of interpolation, adaptive thresholding, and blob detection techniques to monitor the number and position of individuals within the detection area. When two or more individuals are in close proximity, causing their thermal regions to merge, the system employs a segmentation algorithm to effectively separate the heat signatures and maintain accurate localization.

Continuing from our previous research, this work extends the study presented in [8]. In this expanded version, we have removed the previously developed Adaptive Background Estimation and Subtraction (ABES) algorithm and incorporated its core concepts into an enhanced adaptive thresholding technique, thereby streamlining the foreground-background separation process and improving system efficiency. In addition, the integration of an adaptive sliding window allows the system to adapt more effectively to varying human movement speeds, thereby enhancing the stability of the detection process.

The remainder of this paper is organized as follows: Section 2 presents the background, motivation, and a review of work relevant to the proposed method. Section 3 details the human occupancy monitoring and location detection methods used in this study. Section 4 presents the experimental setup and analysis of results. Section 5 concludes the paper.

## 2. Background Knowledge

Human occupancy monitoring and positioning in various environments is a crucial area of research in computer vision [9,10], with wide-ranging applications such as smart building management, security systems, and efficient energy use. Traditional methods relying on visible-spectrum cameras face significant challenges, including privacy concerns and sensitivity to varying lighting conditions. An alternative approach to address these issues is the use of infrared thermal imaging, which detects the heat emitted by objects. This section provides an overview of the principles of thermal imaging, the challenges associated with its use, and the motivation for this research.

Infrared thermal imaging operates by detecting infrared radiation emitted by objects as a function of their temperature. Infrared thermal sensors, such as the AMG8833 [7] used in this study, convert this detected radiation into electrical signals, which are subsequently processed to generate thermal images. These images show the temperature distribution within the scene, where warmer objects, such as human bodies, appear as distinct heat signatures against cooler backgrounds.

One of the primary advantages of thermal imaging is its ability to function effectively in complete darkness and through obstructive conditions such as smoke or fog, which would hinder visible spectrum cameras. Furthermore, since thermal imaging captures heat patterns rather than visual details, it naturally protects individual privacy, making it ideal for use in sensitive environments like residential spaces or workplaces where privacy is a concern. However, thermal imaging also presents several challenges. The low resolution of cost-effective thermal sensors, such as the AMG8833 with its 8 × 8 pixel array, limits the detail that can be captured, which complicates the accurate detection and positioning of human occupancy, particularly in scenarios with multiple occupants where heat signatures may overlap.

Additionally, the dynamic nature of thermal environments poses further challenges. Human movement and fluctuations in ambient temperature can cause significant changes in the thermal image, making it difficult to maintain a consistent model of occupancy. Noise and artifacts introduced by the sensor add another layer of complexity to the processing of thermal images, necessitating advanced techniques to accurately monitor and position human occupancy in real time.

### 2.1. Motivation

The motivation for this research arises from the pressing need to develop a reliable and privacy-preserving system for real-time human occupancy monitoring and positioning across diverse environments. Traditional vision-based monitoring systems, such as high-resolution cameras, frequently encounter difficulties in sub-optimal lighting conditions and other external factors, which can lead to degraded image quality and reduced accuracy in occupancy detection. Additionally, these systems pose significant privacy concerns due to the detailed visual data they collect, which can be susceptible to misuse.

Our focus is particularly on environments such as healthcare facilities and nursing homes, where monitoring human occupancy is critical to ensuring the safety and well-being of individuals with limited mobility. In these settings, an effective monitoring system is vital for detecting unusual patterns of movement or occupancy and facilitating timely interventions. Furthermore, in high-risk areas like industrial sites or hazardous work environments, accurate monitoring and positioning of human occupancy are essential for preventing accidents, maintaining compliance with safety protocols, and enhancing overall safety management.

To show the novelty of the system, we list in Table 1 several methods commonly used in Japanese nursing homes. In LASHIC, sensors can be used to detect the presence of a person (e.g., whether they are in the room, moving, etc.), but it is difficult to use them for tracking. In mamoAI and Neos+Care, motion sensors and cameras are used together for fall detection. Although images are mosaicked in mamoAI and silhouette is extracted in Neos+Care, leakage of the original image data remains a potential risk.

To address these challenges, we propose a system that leverages low-resolution infrared thermal sensors combined with edge computing capabilities. This approach not only preserves privacy by capturing thermal images instead of detailed visual information but also ensures reliable performance across various environmental conditions. By processing data locally on a Raspberry Pi, our system reduces latency and enables real-time occupancy monitoring and positioning, making it highly suitable for deployment in both healthcare and high-risk environments.

### 2.2. Research Objective

The primary objective of this research is to develop a real-time human occupancy monitoring and positioning system that can operate effectively and stably in diverse indoor environments. This system is designed to address challenges such as dynamic human activities, varying environmental conditions, and overlapping heat sources. Our approach integrates multiple image processing techniques and adaptive algorithms to achieve high detection accuracy, stable tracking, and precise positioning. Specifically, the key contributions of this research are an improved adaptive thresholding technique, a robust image segmentation method, and an adaptive sliding window strategy for tracking. By focusing on these areas, the proposed system aims to improve robustness, flexibility, and adaptability, thus contributing to the broader field of human occupancy monitoring and smart space management.

#### 2.2.1. Adaptive Thresholding

Accurate foreground segmentation in thermal imaging is a critical step, as thermal data are often susceptible to background noise and residual heat from previous human movement. In our prior research, the Adaptive Background Estimation and Subtraction (ABES) algorithm was used to perform an initial separation of the foreground and background. The ABES algorithm utilized an exponentially weighted moving average (EWMA) to incrementally update the background model, enabling the system to adapt to dynamic environmental changes. However, the presence of residual heat after human movement often led to errors in background estimation, thus necessitating the use of an Adaptive Thresholding technique to refine the segmentation.

In this study, the computational framework of ABES has been integrated into the Adaptive Thresholding technique, allowing the system to directly distinguish between foreground and background without requiring separate, iterative background updates. This integration not only reduces computational complexity but also improves the robustness of the segmentation process by eliminating residual heat artifacts more effectively. The adaptive threshold is determined using statistical properties of the thermal image data, such as median and Median Absolute Deviation (MAD), allowing it to dynamically adjust to varying scene conditions. Consequently, the improved adaptive thresholding approach significantly strengthens the system’s capability to accurately segment human presence under challenging thermal scenarios.

#### 2.2.2. Image Segmentation

Isolating human heat signatures in thermal images is complex, especially when multiple people are present or when the proximity between them leads to the merging of thermal regions. The segmentation process in our system is performed in two stages. Initially, Connected Component Labeling (CCL) is used to identify and label contiguous regions representing individual human heat signatures. Each labeled region is treated as a separate object, allowing for initial detection and rough positioning.

When multiple human heat sources merge due to proximity or motion, CCL alone may misidentify them as a single entity. To address this issue, we integrate the watershed algorithm, which interprets the intensity values in the image as a topographic surface. The algorithm treats high-intensity regions as ‘peaks’ and low-intensity areas as ‘valleys’. By applying watershed segmentation, overlapping regions are effectively separated, preserving individual human positions even when physical boundaries are unclear. This combined segmentation approach ensures precise identification and localization of each human presence, even in complex and crowded environments.

#### 2.2.3. Adaptive Sliding Window

Human movements in indoor environments can vary significantly, with sudden changes in speed and direction occurring sporadically. A fixed-size sliding window is often inadequate for handling such dynamic scenarios, as it lacks the flexibility to adapt to varying motion patterns. To address this issue, we implement an Adaptive Sliding Window mechanism, which dynamically adjusts its size based on the detected speed of human targets, allowing the system to respond more effectively to rapid changes in movement.

The primary goal of the Adaptive Sliding Window is to provide a flexible mechanism for tracking that maintains stability during normal, slow-moving conditions and enhances responsiveness when sudden movements are detected. The window size dynamically expands or contracts according to the calculated speed: when high-speed movements are observed, the window size is reduced to minimize the impact of abrupt position changes, thereby maintaining tracking accuracy. Conversely, when the movement slows down, the window size increases to smooth out minor fluctuations, avoiding unnecessary oscillations in the detected positions.

To prevent the Adaptive Sliding Window from interfering with stable and low-speed targets, a speed threshold is incorporated into the mechanism. When the detected speed is below this threshold, the system bypasses the adaptive adjustments and directly tracks the target’s position, ensuring precise localization without introducing any additional smoothing effects. This threshold-based activation ensures that the Adaptive Sliding Window only takes effect under exceptional scenarios, such as when human movement speed suddenly increases.

### 2.3. Related Work

The AMG8833 is a widely used infrared thermal imaging sensor with low power consumption and high sensitivity [7]. Its application spans various domains, such as body temperature detection, especially during the COVID-19 pandemic [14,15,16], where it provided a non-contact alternative to traditional methods. However, many of these studies focus on simple temperature measurements rather than complex human detection tasks.

In addition to basic applications like body temperature detection, several studies have explored using AMG8833 for advanced safety applications. For example, ref. [17] developed a fall detection system that leverages thermal imaging to monitor sudden movements. However, this system’s reliability may be affected when individuals remain inactive for extended periods, as human heat can blend into the background.

In industrial applications, AMG8833 has been integrated into autonomous robots for environmental inspection and rescue operations [18]. This research highlights the AMG8833’s potential in low-visibility settings, such as mining environments. However, in narrow and enclosed spaces like mines, overlapping human thermal signatures can lead to detection errors.

To overcome the resolution challenges, bicubic interpolation is commonly applied. Ref. [19] discusses the use of bicubic interpolation, which provides a more refined reconstruction of image material than the relatively simple bilinear interpolation. Some approaches using adaptive thresholding adjust thresholds based on local image statistics, ref. [20] taking into account spatial lighting changes, suitable for real-time image processing. Ref. [21] introduced an adaptive thresholding approach, which calculates the global background heat distribution and sets the threshold at 60% of the maximum value. This method, while simple, may struggle under specific conditions, such as when the background temperature is low, leading to misclassifications of moving humans as background heat sources. Further, ref. [22] proposed an iterative thresholding algorithm that dynamically adjusts based on Gaussian-filtered temperature values. Although effective, this algorithm may require multiple iterations, which could affect real-time processing in environments with many people.

Additionally, recent work by [23] used a convolutional encoder-decoder network for human detection, achieving 98.43% accuracy with Adaptive Boosting (AdaBoost) across different sensor configurations. Despite these advancements, other challenges remain, particularly in distinguishing closely positioned individuals. Ref. [24] tackled this issue by applying connected component labeling and blob filtering to thermal data, accurately segmenting densely standing individuals. The method, though effective, still relies heavily on the sensor layout and may face limitations in highly dynamic environments.

Although various methods have improved the use of low-resolution thermal sensors, challenges remain in accurately detecting and distinguishing human objects in dynamic environments. Our approach builds on these methods, integrating bicubic interpolation, adaptive thresholding, connected component labeling, watershed algorithm, and the adaptive sliding window. This combination addresses key limitations, providing a more robust solution for real-time human detection and positioning in dynamic indoor environments.

## 3. Methodology

In this article, we propose an integrated framework to enhance the accuracy of human occupancy detection and positioning in indoor environments using thermal imaging data. Specifically, our system applies bicubic interpolation to smooth the low-resolution thermal data captured by the AMG8833 sensor (Panasonic, Kadoma, Japan), followed by an improved adaptive thresholding approach to separate the foreground from the background. In this section, we provide a detailed description of the architecture and components of our proposed system. The proposed framework consists of four key functional modules: data preprocessing with bicubic interpolation, adaptive thresholding, blob detection using connected component analysis, and segmentation refinement via the watershed algorithm. Finally, a dynamic adjustment strategy for sliding windows is introduced to enhance the accuracy of human movement detection.

### 3.1. Data Collection

The data collection process in this study is carried out using the AMG8833 infrared thermal imaging sensor, which has been widely utilized in applications such as human detection and environment monitoring [25]. This sensor captures low-resolution thermal data in a grid format of 8 × 8 pixels, where each pixel represents the temperature value detected within a given field of view. The sensor operates within a temperature range of 0 °C to 80 °C, with a maximum frame rate of up to 10 Hz. Given the low resolution of the collected thermal data, additional preprocessing steps are required to enhance the quality of the data for human detection and positioning.

The AMG8833 communicates with the data processing system through an I2C (Inter-Integrated Circuit) interface, using a slave address of 0 × 69 for data transmission. The communication protocol allows the sensor to be easily integrated with systems such as Raspberry Pi4 for real-time data acquisition. During data collection, the sensor continuously captures the thermal heat distribution in the surrounding environment, transmitting the temperature values from each of the 64 pixels over the I2C bus at regular intervals. The key technical specifications of the AMG8833 sensor, including its pixel resolution, detection range, and temperature detection range, are summarized in Table 2.

Figure 1 illustrates a typical output of the AMG8833 sensor, where the thermal data are visualized as an 8 × 8 heatmap, with lighter colors indicating higher temperatures and darker colors representing lower temperatures. These upsampled data are then processed through subsequent steps in our framework, such as adaptive thresholding and blob detection, to accurately detect and track human presence in indoor environments. The temperature scale on the right-hand side provides a reference for interpreting the values displayed in the heatmap, with a range from approximately 23.5 °C to 27.5 °C in this specific output. The variation in color shades reflects the temperature gradient across the captured area, which can correspond to various objects or individuals in the environment that emit heat.

### 3.2. Interpolation

Previous studies [21] show that thermal data from the AMG8833 sensor are often affected by environmental noise and body temperature fluctuations, reducing image analysis accuracy. Pre-processing is essential to improve performance and reliability.

To address these challenges, our system applies interpolation to refine thermal data and enhance resolution. The raw 8 × 8 grid of thermal readings from AMG8833 is interpolated into a higher resolution matrix, resulting in smoother thermal images. This step improves spatial resolution and provides a more detailed heat distribution, enhancing downstream detection accuracy. We compared three interpolation methods: nearest-neighbor, bilinear, and bicubic interpolation. In the following, each method is discussed in detail.

#### 3.2.1. Nearest-Neighbor Interpolation

Nearest-neighbor interpolation assigns a new pixel’s value based on the nearest known pixel, preserving sharp transitions but often resulting in blocky or jagged edges, especially in low-resolution images. Although computationally fast, it tends to produce visual artifacts, such as jagged edges, making it unsuitable for applications requiring smooth gradients or fine detail.

#### 3.2.2. Bilinear Interpolation

Bilinear interpolation estimates the new pixel value by linearly interpolating the four nearest pixels, producing smoother transitions than nearest-neighbor. This method reduces jagged edges but may still cause slight blurring, especially in areas with sharp temperature changes. It offers a balance between computational efficiency and image quality.

#### 3.2.3. Bicubic Interpolation

Bicubic interpolation uses 16 surrounding pixels to compute a new pixel’s value, achieving smoother transitions and finer details. It delivers superior image quality compared to nearest-neighbor and bilinear methods, making it ideal for applications demanding high visual clarity. However, it requires significantly more computational resources.

Table 3 highlights the advantages of bicubic interpolation, which offers superior visual quality with fine details and smooth gradients. Although more computationally intensive, it excels in high-resolution tracking and precise segmentation, making it ideal for applications requiring accurate thermal region differentiation.

Figure 2 demonstrates the impact of bicubic interpolation on an 8 × 8 heatmap from the AMG8833 sensor, clearly showing its ability to distinguish closely adjacent thermal regions, supporting its use in human tracking and thermal segmentation tasks.

### 3.3. Adaptive Threshold

In our previous study [8], the Adaptive Background Estimation and Subtraction (ABES) algorithm primarily utilized the Exponentially Weighted Moving Average (EWMA) method to process each frame of data. By adjusting the value of each pixel incrementally over successive frames, the system was able to adapt to changes in the environment based on historical data. This approach allowed the system to dynamically update the background model, thereby improving the accuracy of foreground detection. However, the ABES algorithm has certain limitations. First, ABES introduces significant computational complexity, as it requires continuous background updates for each frame of data. This process increases the computational load, particularly in scenarios that demand high-frequency data processing, potentially affecting real-time performance. Additionally, the multi-step process of ABES, including EWMA updates and subsequent foreground segmentation, introduces extra processing time and steps. Ultimately, although ABES effectively separates foreground and background, these steps become unnecessary after applying bicubic interpolation, as the thermal map is eventually binarized for further processing.

To address the aforementioned issues, we propose an improved adaptive thresholding method in this study to replace the ABES algorithm. In the original design, the adaptive threshold was calculated based on the mean and standard deviation. However, this approach has some drawbacks, particularly in cases involving background noise or residual human heat. The mean is highly susceptible to extreme values, and during rapid background changes or abnormal data fluctuations, it can deviate from the central tendency, leading to inaccurate threshold settings. The standard deviation is also sensitive to data variation, which can result in unstable detection, especially when there are significant fluctuations in the thermal map. These issues become particularly evident in dynamic environments, potentially leading to false positives or missed detections. The formulas in our previous study [8] are as follows:(1)Threshold=Mean+Sensitivity×SD
(2)$$Sensitivity=\frac{sensitivity\_base}{\sqrt{n+1}}$$

Here, *n* represents the number of detected individuals, and *sensitivity*_*base* is the system’s base sensitivity constant, we introduce +1 to avoid a zero denominator when *n* = 0, ensuring that the formula remains valid even when no individuals are detected. This method effectively maintains sensitivity stability when the number of detected individuals is low. However, as the number of detected individuals increases, this method exhibits significant limitations. The rate of change in the square root function gradually slows as the values grow larger, causing the sensitivity to decrease with an increase in the number of individuals. This results in significantly reduced detection sensitivity in environments with large groups of people, thereby increasing the risk of false detection.

To improve stability, our revised adaptive thresholding method uses the *Median* and median absolute deviation (*MAD*) to calculate the threshold. The median, as a more robust measure of central tendency, effectively reduces the impact of outliers on the threshold, enhancing system stability in environments with high background noise. *MAD*, as a complementary measure to the median, more accurately reflects data dispersion and is less affected by extreme data points. This improvement ensures that the system maintains stable detection performance even in highly fluctuating scenarios, enabling effective separation of foreground and background. The formulas are as follows:(3)Threshold=Median+Sensitivity×MAD
(4)Sensitivity=sensitivity_base×log(1+n)×temp_compensation
The log(1+n) term gradually increases sensitivity as the number of detected individuals rises. The *temp_compensation* factor is an adjustment based on the ambient temperature, designed to account for the effects of seasonal or environmental temperature changes on detection sensitivity. The temperature compensation factor is calculated as follows:(5)$$temp\_compensation=\left\{\begin{array}{cc} 1+\frac{T_{ref1}-ambient\_temp}{C}, & \mathrm{if}\;ambient\_temp<T_{ref1}\\ 1-\frac{ambient\_temp-T_{ref2}}{C}, & \mathrm{if}\;ambient\_temp>T_{ref2}\\ 1, & \mathrm{if}\;T_{ref1}\leq ambient\_temp\leq T_{ref2} \end{array}\right.$$
In this equation, $T_{\mathrm{ref}1}$ and $T_{\mathrm{ref}2}$ represent the lower and upper reference temperatures, respectively, while *C* is the compensation constant that controls the degree of sensitivity adjustment based on the ambient temperature. The value of *C* is selected to ensure that the compensation factor is neither too aggressive nor too minimal, providing a balance that allows the system to respond to changes in environmental temperature without causing excessive fluctuations in the detection sensitivity. The choice of *C* can be tuned depending on the specific environmental conditions or the desired behavior of the system.

For typical indoor environments, the temperature range is usually between $T_{\mathrm{ref}1}$ = 22 °C and $T_{\mathrm{ref}2}$ = 28 °C, as human thermal signatures are most stable within this range. When the ambient temperature drops below $T_{\mathrm{ref}1}$, the system increases the sensitivity to distinguish human heat signatures from the cooler background. Conversely, when the temperature rises above $T_{\mathrm{ref}2}$, the system decreases the sensitivity to avoid over-detecting heat from the background.

The revised method is directly applied to the thermal map after bicubic interpolation. The adaptive thresholding technique separates the foreground from the background and converts the thermal map into a binarized image. This process eliminates the need for per-frame background model updates, and by incorporating the median and MAD-based adaptive thresholding, it replaces the key steps in the ABES algorithm. This method not only simplifies the processing workflow but also enhances system efficiency, effectively addressing the redundant steps in the ABES algorithm. Table 4 presents a comparison between the old method and the new method.

### 3.4. Connected Component Labeling

Connected Component Labeling (CCL) is a fundamental image processing technique used to identify and label connected regions in binary images, either with 4-connectivity or 8-connectivity. In 4-connectivity, each pixel is connected to its four neighboring pixels (up, down, left, and right). In 8-connectivity, a pixel is connected not only to its four neighboring pixels but also to its four diagonal neighbors, as illustrated in Figure 3. In our system, after applying the adaptive thresholding method, the thermal image is binarized, with regions corresponding to detected human bodies. CCL is then employed to group pixels that are connected and represent distinct entities in the thermal image.

The goal of using CCL is to ensure that each distinct thermal signature is uniquely labeled, which facilitates subsequent analysis and tracking of detected individuals. By effectively labeling connected regions, CCL allows for the precise counting of individuals and enables further processing steps, such as tracking movement or analyzing group behaviors.

In our implementation, we use an 8-connectivity approach, meaning that a pixel is considered connected to its neighbors if they share a side or corner. This ensures that even diagonally connected regions are treated as part of the same entity, which improves the robustness of human detection in cases where individuals may be partially overlapping. The output of CCL is a set of labeled regions, each assigned a unique identifier, which are then used for further processing and analysis. Figure 4 shows the process of applying connected component labeling to the thermal map, starting from raw thermal data, followed by interpolation, adaptive thresholding, and finally, identifying distinct human regions using CCL. The labeled regions are displayed in the binary image, where each identified individual is enclosed within a bounding box and assigned a unique identifier.

### 3.5. Watershed Algorithm

A reliable segmentation algorithm is required to address a common issue observed in thermal imaging, where the thermal signatures of multiple individuals standing in close proximity may merge. In this study, the watershed algorithm was selected to effectively mitigate this problem. As illustrated in Figure 5, when two individuals are in close proximity, their thermal signatures may merge into a single region, causing the system to incorrectly identify them as a single entity. To address this issue, the watershed algorithm is introduced to accurately separate these overlapping regions, ensuring precise detection and labeling of each individual.

The watershed algorithm is used to effectively segment overlapping regions by transforming the binary image into a topographic representation that allows distinct boundaries to be detected. Since thermal images are converted to binary images through adaptive thresholding, they need to be transformed into a topographic image via the distance transform. The distance transform, using the Euclidean distance, converts each foreground pixel to a value representing its distance to the nearest background pixel. The resulting image from the distance transform can be viewed as a topographic surface, where higher values indicate pixels farther from the background and lower values indicate pixels closer to the background, as shown in Figure 6.

It operates by treating this transformed image as a topographic surface, where markers are generated by identifying local maxima within each region, representing likely center points. These markers are placed in the central regions and serve as the starting points for the flooding process. During this process, regions expand outwards until they meet boundaries formed by neighboring regions or the edge of the image. These boundaries are used to separate overlapping regions, effectively isolating each individual.

Figure 7 illustrates the application of the watershed algorithm to separate thermal signatures with closely positioned positions. In Image (A), because the distance between the markers is less than the defined threshold, the watershed algorithm is applied. Image (B) shows the resulting segmented thermal image after applying this algorithm, effectively isolating the thermal signatures of each individual.

### 3.6. Adaptive Sliding Window

The Adaptive Sliding Window technique is designed to enhance the stability in human positioning, particularly during scenarios where movement speed varies significantly. This mechanism allows the system to adaptively adjust its temporal window size based on the detected speed of human movement, thereby optimizing tracking precision only when needed.

The sliding window is initialized with both a minimum and maximum window size. The minimum window size allows for responsiveness to rapid movement, while the maximum size ensures greater temporal smoothing during slower movements. The system uses a speed threshold to determine when the adaptive adjustment should be activated. Only when the detected speed exceeds the threshold is the adaptive sliding window activated, allowing for window size adjustments and smoothing of position estimates. When the movement speed is below the threshold, the adaptive sliding window is disabled, and the system returns directly to the current position without applying any smoothing. Figure 8 visually demonstrates how the window size dynamically changes over time as speed varies, emphasizing the conditional activation of the adaptive mechanism.

To achieve this, the Euclidean distance between consecutive positions is calculated to represent movement speed. If this calculated speed exceeds the threshold, the sliding window adjusts its size accordingly to maintain optimal tracking accuracy. The adaptive sliding window retains a history of recent positions and calculates an average to determine a smoothed position estimate. By applying this approach, the system achieves a balance between responsiveness and stability, which is crucial for scenarios with varying human activity.

In practical deployment, this sliding window mechanism is particularly useful in environments where human movement can alternate between dynamic and stationary states. When the system detects a sudden increase in speed, the window size is reduced, allowing the algorithm to quickly respond to changes. When the speed is below the threshold, the window mechanism is disabled, focusing on accuracy and reducing computational load by avoiding unnecessary smoothing.

### 3.7. Summary of Methodology

To summarize, the proposed framework enhances human occupancy detection and positioning by improving image resolution, noise resilience, and segmentation accuracy through the integration of classical image processing techniques with key enhancements. The system begins with bicubic interpolation to upsample 8 × 8 thermal images, providing enhanced spatial resolution for subsequent analysis. The proposed adaptive thresholding method, based on median and MAD, offers better noise resilience and segmentation accuracy compared to prior thresholding approaches, such as those relying on mean and standard deviation. Connected component labeling (CCL) is applied to identify and label distinct thermal regions corresponding to human figures, while the watershed algorithm is utilized to separate overlapping thermal signatures. Finally, an adaptive sliding window strategy is introduced to improve detection stability in dynamic environments with human movement. These enhancements collectively achieve more robust, accurate, and computationally efficient human tracking in thermal images. 

### 3.8. Analysis of Computational Cost

To evaluate the system’s computational complexity, we analyzed the time complexity of each method and its role in ensuring real-time performance on the Raspberry Pi 4 (Raspberry Pi Trading Ltd., Cambridge, UK). Table 5 is a summary of each method and its theoretic (worst case) time complexity.

## 4. Experimental Result

### 4.1. Experimental Setting

The experimental setup for this research involved using a Raspberry Pi 4 Model B with 8 GB RAM as the main processing unit. The Raspberry Pi 4 was selected for its edge computing capabilities, which allow data processing at the source, mitigating potential issues related to network latency. The Raspberry Pi 4 was mounted on the ceiling at an approximate height of 2.7 m from the ground. The AMG8833 infrared thermal sensor was connected to the Raspberry Pi, serving as the primary device for detecting human presence. Experimental measurements indicated that, when placed at a height of 2.7 m, the AMG8833 sensor effectively covered a detection area of approximately 2 × 2 m on the ground, forming a square coverage zone, as shown in Figure 9. Adjacent to the thermal sensor, a camera module was installed, which also interfaced with the Raspberry Pi. The purpose of the camera was solely to validate the thermal sensor’s performance in terms of detecting human presence and accurately determining positioning. The complete system configuration is shown in Figure 10, and the pin diagram is shown in Figure 11.

### 4.2. Experimental Results

The experimental results, including video data, are available on GitHub (see https://reurl.cc/6doV96 (accessed on 22 December 2024) (GitHub Repository)). The video data demonstrate the accuracy of the estimated locations of the subjects compared to their true locations, showing consistent results regardless of the subjects’ positions.

The experimental results included a series of tests involving one, two, and three individuals moving within the detection area to evaluate the system’s performance in various real-world scenarios. The system’s ability to accurately detect and track multiple individuals was tested under different conditions, with the results captured and analyzed in detail.

Figure 12 illustrates the result when an individual was moving within the environment. The system was able to accurately detect and track the individual, demonstrating the reliability of the adaptive thresholding technique in maintaining consistent detection. In particular, the individual temperature detected was approximately 27.57 °C, which is significantly lower than the expected human body surface temperature of 36 °C to 37 °C. The discrepancy is due to the AMG8833 sensor’s installation height and wide field of view, which cause averaging of human thermal signatures with the cooler surroundings and attenuation of infrared radiation. These limitations result in a detected temperature that is lower than the actual temperature of the body surface.

For two individuals, the results are shown in Figure 13. The system successfully detected both individuals and was able to distinguish their movements clearly, highlighting the robustness of the adaptive techniques used.

Similarly, Figure 14 shows three individuals moving within the detection area. The system continued to perform well, accurately identifying and tracking each individual without significant errors, even as the number of people increased.

In addition to testing during standard lighting conditions, experiments were conducted in complete darkness to verify the system’s ability to function effectively regardless of ambient lighting. As shown in Figure 15, the system demonstrated excellent performance in detecting human presence and accurately determining positions in the absence of light. Since the AMG8833 infrared thermal sensor detects heat signatures rather than relying on visible light, it enabled consistent detection, proving the system’s capability for 24-h operation regardless of whether it was day or night.

The watershed algorithm was used to address scenarios where individuals came into close proximity, resulting in merged thermal signatures. When thermal regions overlapped, the watershed algorithm effectively segmented them, allowing the system to accurately detect distinct individuals, as shown in Figure 16. This capability ensures that the system maintains its reliability even in situations where human subjects are in close proximity, thereby enhancing the robustness of the detection process.

### 4.3. The Measured Computational Cost

To validate the low latency and real-time performance, we measured the actual processing speed on the Raspberry Pi 4. The computational cost for each frame is calculated in terms of floating-point operations (FLOPs). The results are as follows:FLOPs per frame: 159,744 FLOPsFLOPs per second (at 10 FPS): 159,744 × 10 = 1.6 MFLOPS

The Raspberry Pi 4 provides a peak computational capacity of 13 GFLOPS. Since the proposed system requires around 1.6 MFLOPS, real-time processing can be guaranteed.

Additionally, the average processing time per frame is approximately 2 milliseconds, ensuring the system can operate efficiently at real-time speeds (10 FPS). This lightweight computational demand allows the system to provide continuous, real-time detection on low-cost, low-power hardware, such as the Raspberry Pi 4, making it suitable for on-site deployment in healthcare and nursing home environments where system responsiveness is critical.

## 5. Conclusions

In this study, we presented a human occupancy monitoring and positioning system that uses an infrared thermal array sensor and various image processing techniques. The system demonstrated its effectiveness in detecting and tracking human presence under different indoor conditions, including standard lighting and complete darkness, without compromising privacy. By integrating adaptive thresholding and the watershed algorithm, we achieved robust segmentation of thermal regions, even when multiple individuals came into close proximity, thus enhancing the accuracy and reliability of human detection.

The implementation of the speed-responsive adaptive sliding window allowed for effective tracking by dynamically adjusting to changes in movement speed, providing a balance between tracking stability and responsiveness. This feature was particularly useful in environments with varied human activities, ensuring accurate positioning across different scenarios.

Although the current system performed well under controlled experimental conditions, further research could explore the system’s adaptability to more complex real-world environments with a higher density of occupants or increased levels of noise. Additionally, optimizing computational efficiency for deployment in resource-constrained edge computing devices could be a valuable direction for future work.

Overall, this research contributes to the advancement of privacy-preserving, real-time human occupancy monitoring solutions, with potential applications in smart building management, healthcare, and security systems.

## Figures and Tables

**Figure 1 sensors-25-00129-f001:**
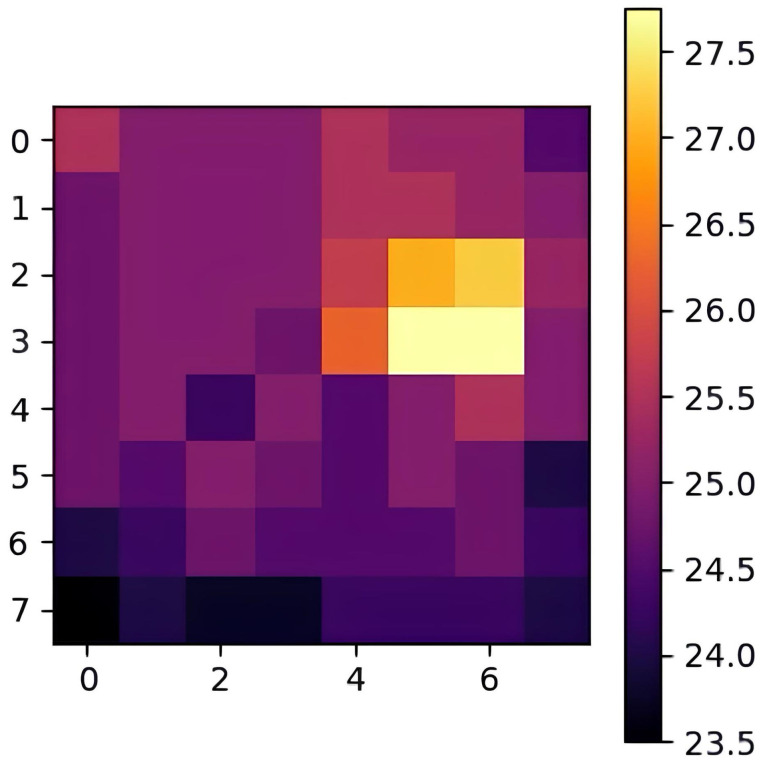
Original heatmap from AMG8833.

**Figure 2 sensors-25-00129-f002:**
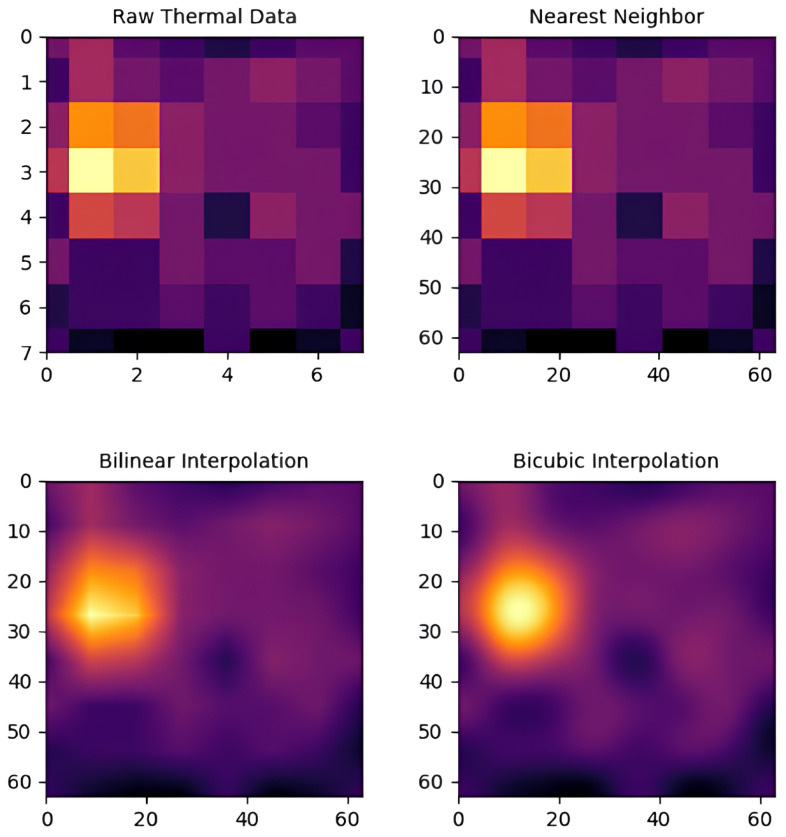
Different interpolation methods applied to original heatmaps.

**Figure 3 sensors-25-00129-f003:**
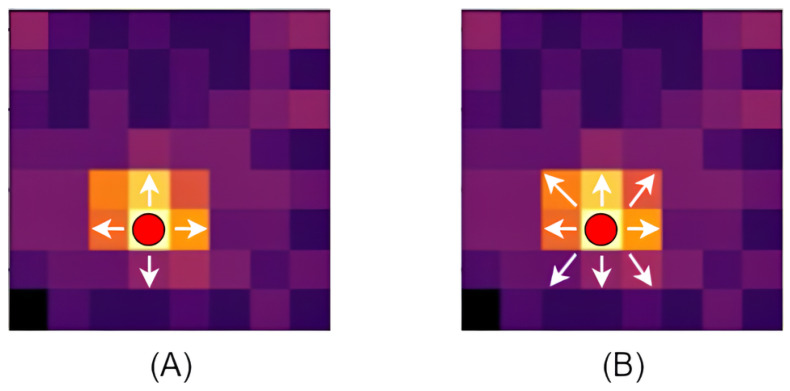
(**A**) 4-connectivity. (**B**) 8-connectivity.

**Figure 4 sensors-25-00129-f004:**
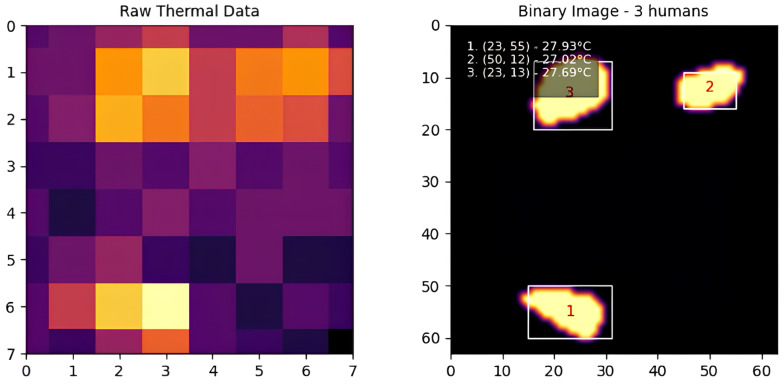
Connected component labeling results.

**Figure 5 sensors-25-00129-f005:**
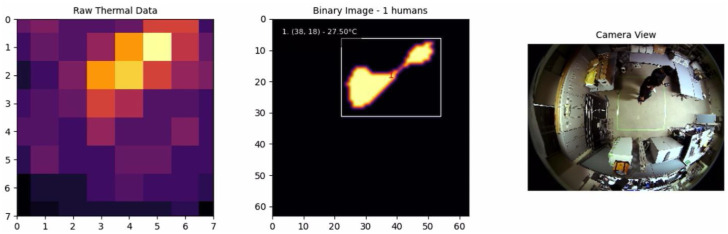
Thermal merger due to close proximity.

**Figure 6 sensors-25-00129-f006:**
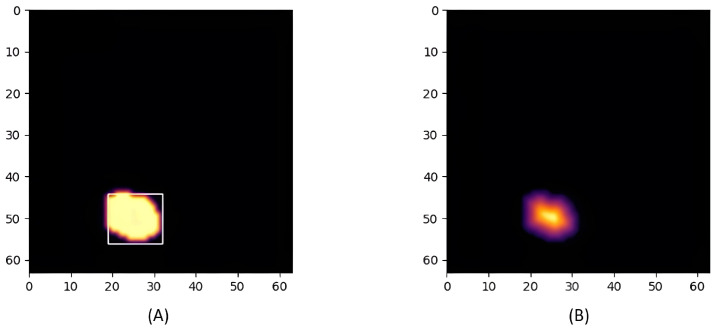
(**A**) Binary Image. (**B**) Topographic Image.

**Figure 7 sensors-25-00129-f007:**
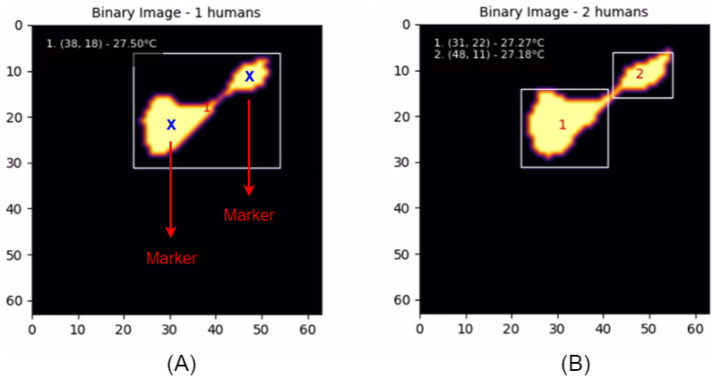
(**A**) Detection Error. (**B**) Correct Detection.

**Figure 8 sensors-25-00129-f008:**
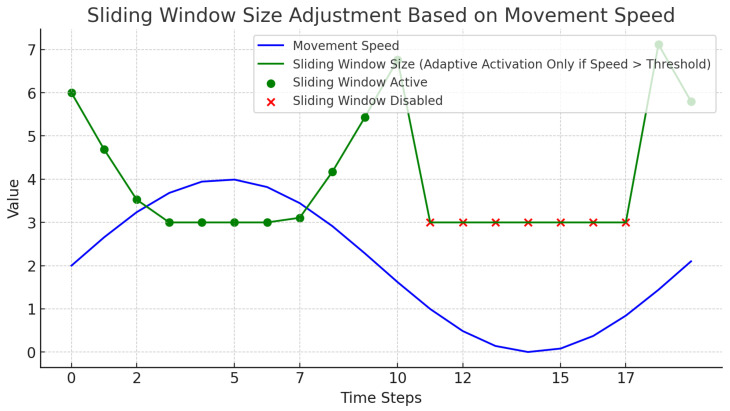
Sliding window size adjustment based on movement speed.

**Figure 9 sensors-25-00129-f009:**
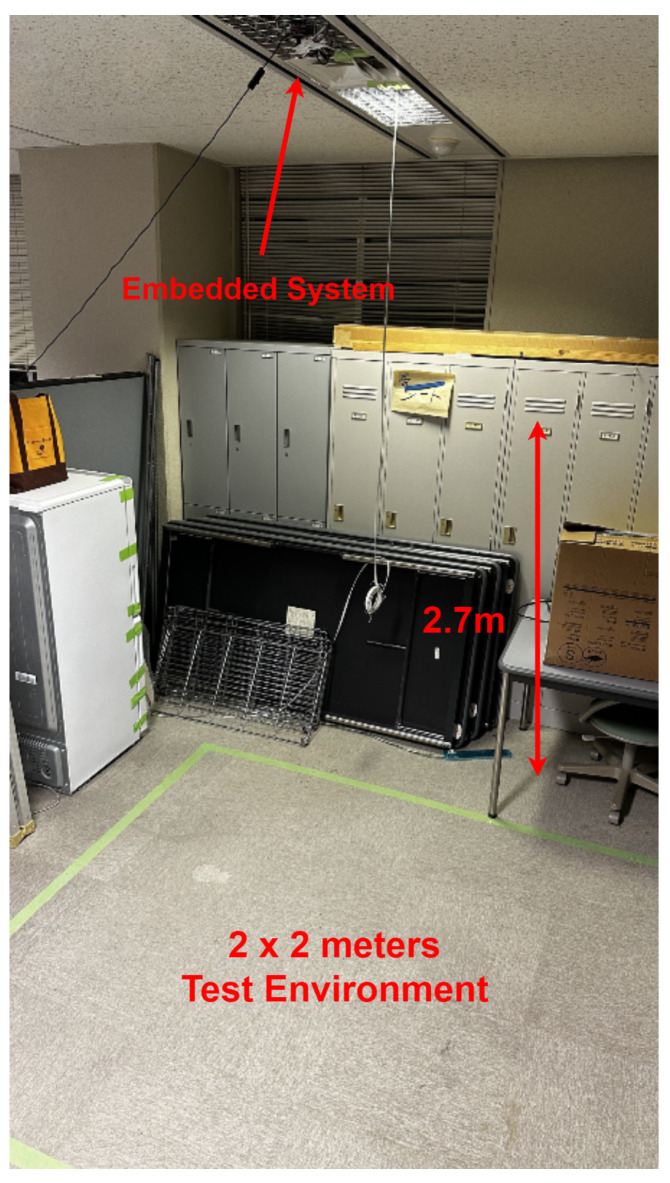
Test environment.

**Figure 10 sensors-25-00129-f010:**
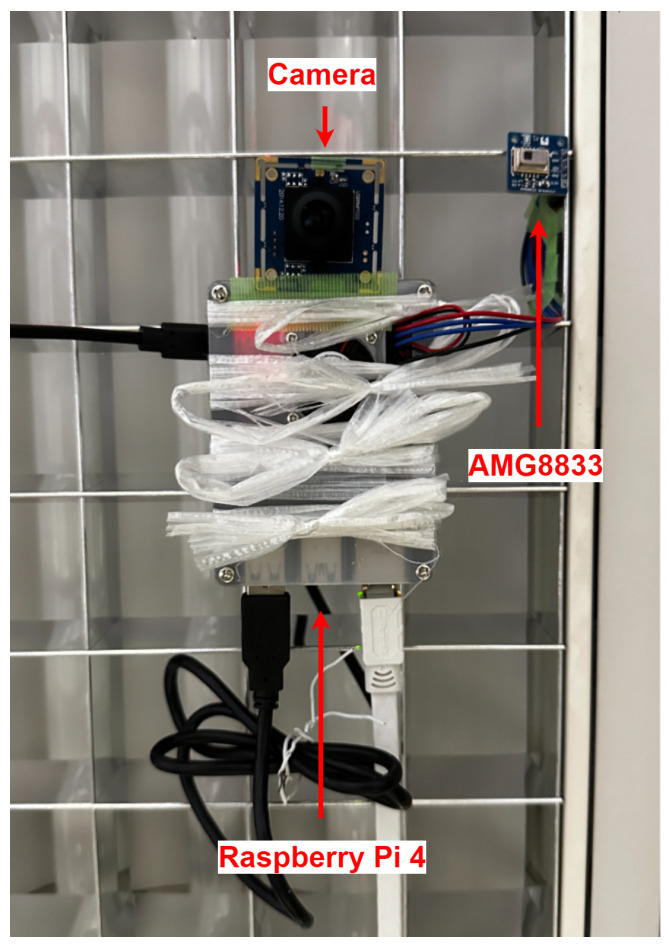
Hardware configuration.

**Figure 11 sensors-25-00129-f011:**
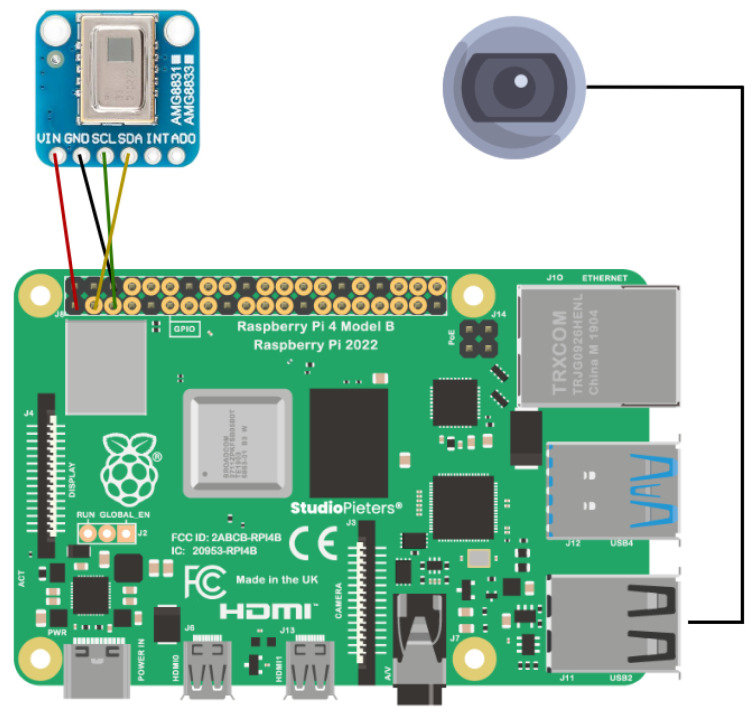
Pin diagram.

**Figure 12 sensors-25-00129-f012:**
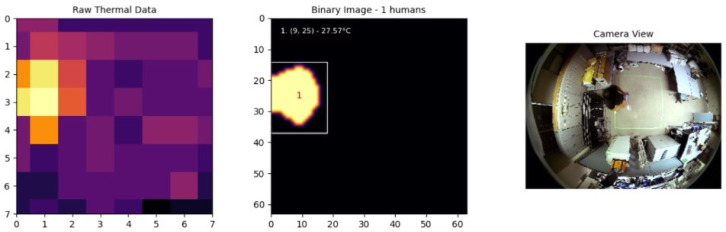
One individual.

**Figure 13 sensors-25-00129-f013:**
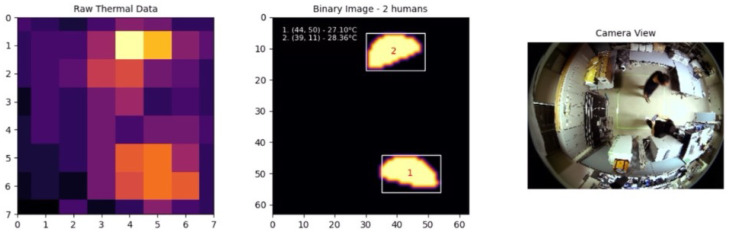
Two individuals.

**Figure 14 sensors-25-00129-f014:**
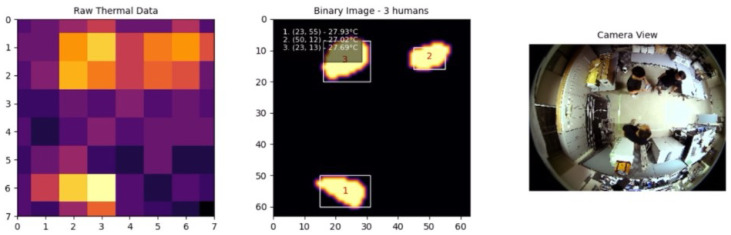
Three individuals.

**Figure 15 sensors-25-00129-f015:**
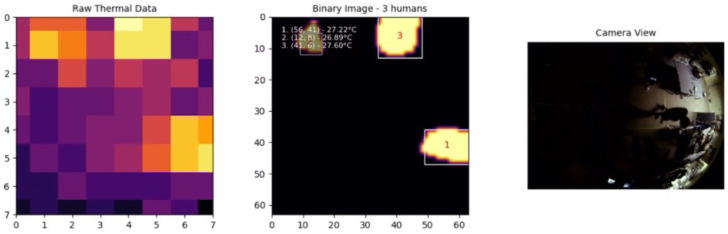
Dark environment.

**Figure 16 sensors-25-00129-f016:**
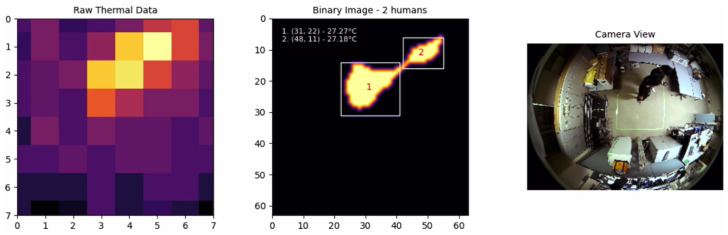
Watershed algorithm applied.

**Table 1 sensors-25-00129-t001:** Comparison with existing systems used in nursing homes.

System	Type	Features	Limitations
LASHIC [11]	Various environmental sensors	Detects light, temperature, pressure, presence of human, etc.	Cannot track human location
mamoAI [12]	Motion sensor + video camera	Fall detection	Privacy concerns
Neos+Care [13]	Motion sensor + video camera	Fall detection	Privacy concerns
Proposed System	Thermal array sensor	Detect location and movement	Can be used only in small private space

**Table 2 sensors-25-00129-t002:** Detail of AMG8833 infrared thermal sensor.

Parameter	Value
Pixel Resolution	8 × 8 (64 pixels)
Detection Range	7 m (23 feet)
Operating Voltage	3.3 V
Maximum frame rate	10 Hz
Communication Protocol	I2C (Inter-Integrated Circuit)
Temperature Detection Range	0 °C to 80 °C (32 °F to 176 °F)

**Table 3 sensors-25-00129-t003:** Comparison of interpolation methods.

Method	Computational Efficiency	Edge Quality	Suitable Application Scenarios
Nearest-Neighbor	High	Jagged edges	Situations requiring fast processing with low image quality requirements
Bilinear	Medium	Some blurring	Medium-resolution scenarios needing basic smoothness
Bicubic	Low	Smooth and detailed	High-quality, detailed applications requiring visual clarity

**Table 4 sensors-25-00129-t004:** Comparison of previous and improved adaptive thresholding methods.

Item	Previous Adaptive Thresholding	Improved Adaptive Thresholding
Calculation Method	Based on mean and standard deviation (sd)	Based on median and median absolute deviation (MAD)
Sensitivity Adjustment	Inverse square root of the number of people	Logarithmic growth of the number of people
Temperature Compensation	None	Dynamic adjustment based on ambient temperature
Background Noise Handling	Relies on standard deviation, making it sensitive to outliers	Median and MAD make it robust to outliers and noise

**Table 5 sensors-25-00129-t005:** Time Complexity Analysis of the Proposed System.

Method	Time Complexity	Explanation
Bicubic Interpolation	$O(M^{2})$	Interpolate an 8 × 8 heatmap to an M × M image.
Adaptive Thresholding	$O(M^{2}\log M)$	Computes median and MAD for adaptive thresholding. The time complexity is dominated by sorting.
Connected Component Labeling (CCL)	$O(M^{2})$	Two-pass scan for connected components in a binary image.
Watershed Algorithm	$O(M^{2})$	Separates overlapping human regions using distance-based segmentation.
Adaptive Sliding Window	O(1)	Updates human positions using a simple weighted average.

## Data Availability

Data are contained within the article.

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
