# Peer review of "Human Occupancy Monitoring and Positioning with Speed-Responsive Adaptive Sliding Window Using an Infrared Thermal Array Sensor"

_sensors, 2024, doi:10.3390/s25010129_

Round 1

Reviewer 1 Report

Comments and Suggestions for Authors

This manuscript proposed a system using Infrared Thermal array sensor with some methods for human occupancy monitoring. Although this study has practical significance, some concerns should be considered.

1. Many methods have been adopted to enhance the stability of the detection process and are deployed in Raspberry Pi 4 Model. The manuscript declares that “our system reduces latency and enables real-time occupancy monitoring”. Thus whether the computational complexity of the whole system can be analyzed, and show the real-time processing speed?

2. Although many methods have been combined in the system, the novelty is hard to estimate. Could the SOTA methods be compared with the proposed methods?

Reviewer 2 Report

Comments and Suggestions for Authors

The manuscript provides a solid foundation for leveraging Infrared Thermal Sensors in human occupancy monitoring, highlighting their advantages over traditional cameras. The proposed methodology, including interpolation, adaptive thresholding, and a dynamic sliding window, is innovative and addresses key challenges like privacy and environmental variability.

However, the research lacks experimental evidence to validate the effectiveness of the proposed system. Further experiments are needed to demonstrate the system's performance in real-world scenarios, particularly under varying environmental conditions and occupancy levels. Providing quantitative results and comparisons with existing methods would significantly enhance the study's credibility and impact.

Please verify the referencing format used for citing previous publications, such as in "[?] highlights how combining...". I am not familiar with this style, but if it adheres to the journal's guidelines, it is acceptable.

Section 3.2 can be condensed for brevity.

Section 3 includes several classical image processing operations that can be summarized concisely.

Additionally, I would like to see more experimental results to strengthen the study's findings.
